# Developing of Lead/Polyurethane Micro/Nano Composite for Nuclear Shielding Novel Supplies: γ-Spectroscopy and FLUKA Simulation Techniques

**DOI:** 10.3390/polym15224416

**Published:** 2023-11-15

**Authors:** Ahmed M. El-Khatib, Mahmoud I. Abbas, Mohamed E. Mahmoud, Mohammed Fayez-Hassan, Mirvat F. Dib, Mamdouh H. Khalil, Ahmed Abd El Aal

**Affiliations:** 1Physics Department, Faculty of Science, Alexandria University, Alexandria 21511, Egypt; mahmoud.abbas@alexu.edu.eg (M.I.A.); mirvatdib2018@gmail.com (M.F.D.); 2Chemistry Department, Faculty of Science, Alexandria University, Ibrahimia, P.O. Box 426, Alexandria 21321, Egypt; memahmoud10@alexu.edu.eg; 3Egyptian Atomic Energy Authority Nuclear Research Center, Experimental Nuclear Physics, Inshas, Cairo 13759, Egypt; mohammed.fayiz@eaea.org.eg; 4Alex Steel, Desert Road Km 21 Merghm, Alexandria 23722, Egypt; mamdouh.hosny@alexsteel.com.eg; 5Alex Form, Desert Road Km 21 Merghm, Alexandria 23722, Egypt

**Keywords:** lead nano composite, thermogravimetric analysis, X-ray diffraction, Fourier transform infrared analysis, nuclear protection, Monte Carlo, FLUKA

## Abstract

In this work, the effect of adding Pb nano/microparticles in polyurethane foams to improve thermo-physical and mechanical properties were investigated. Moreover, an attempt has been made to modify the micron-sized lead metal powder into nanostructured Pb powder using a high-energy ball mill. Two types of fillers were used, the first is Pb in micro scale and the second is Pb in nano scale. A lead/polyurethane nanocomposite is made using the in-situ polymerization process. The different characterization techniques describe the state of the dispersion of fillers in foam. The effects of these additions in the foam were evaluated, Fourier transform infrared spectroscopy (FTIR), scanning electron microscopy (SEM), transmission electron microscopy (TEM), and X-ray diffraction (XRD) have all been used to analyze the morphology and dispersion of lead in polyurethane. The findings demonstrate that lead is uniformly distributed throughout the polyurethane matrix. The compression test demonstrates that the inclusion of lead weakens the compression strength of the nanocomposites in comparison to that of pure polyurethane. The TGA study shows that the enhanced thermal stability is a result of the inclusion of fillers, especially nanofillers. The shielding efficiency has been studied, MAC, LAC, HVL, MFP and Z_eff_ were determined either experimentally or by Monte Carlo calculations. The nuclear radiation shielding properties were simulated by the FLUKA code for the photon energy range of 0.0001–100 MeV.

## 1. Introduction

In recent years, the use of regular or nano polymer compounds as nuclear shielding materials has attracted the attention of many researchers due to their importance [1,2,3,4,5,6,7,8,9,10]. The qualities of nanopolymer compounds are comparable to those of traditional polymers, and they are ecologically benign [11]. Due to polyurethane foam’s (PU) properties as an effective insulating material, polyurethane foam may be modified to improve its properties [12]. Which are commonly used in many industries, including building engineering, automotive, construction, and furniture [13,14] Polyurethane, often known as PU or PUR, is a polymer made of many organic units connected by urethane molecules. Polyurethanes are excellent insulators; they offer many solutions to the challenges of energy conservation and eco-design. The polyurethane sector is continually looking for methods to decrease its environmental effect and is now looking at ways to improve the energy efficiency of production processes and create products that may be utilized to conserve energy, such as building insulation. Sandwich panels are made of a core of rigid polyurethane foam (RPUF), injected between two layers of pre-painted galvanized steel. As a result, they have enhanced properties that no single mono-material could provide, such as low density, high bending resistance, energy absorption, and high load-bearing capacity with a low specific weight. Separately, the construction, marine, automotive, and aerospace sectors may compete with traditional sheet materials using metal-polymer-metal multilayer composite systems (MPM) [15]. Sandwich panels are lighter than mono-metallic sheets, resulting in reduced fuel consumption. Due to the polymer component sandwiched between the metal layers and their low thermal conductivity, they also provide vibration-dampening and thermal insulation benefits [15,16]. Nanocomposites have gained much interest in recent years as they can lead to new and improved properties when compared to their micro- and macro-composite counterparts. Many methods for producing nanoparticles have been documented to synthesize nano-level materials, such as plasma arcing, chemical vapor deposition, electrodeposition, sol-gel synthesis, high-intensity ball milling, etc. [17,18,19,20]. The ball-milling process is often employed as a mechanical co-grinding of originally dissimilar powders in order to create a new, homogeneous powder. Vials, which are cylindrical receptacles used for the mill and hold balls, are used [21]. The materials that may be used as milling tools include steel, agate, tungsten carbide, and others. Since the particles are broken up during the milling process, new, highly reactive surfaces may result [21]. Many research projects have been conducted to investigate the various radiation-shielding materials to shield life and its surroundings from the consequences of radiation exposure by attenuating or absorbing unwanted radiation [22,23,24,25]. The aim of this study is to examine the shielding abilities of a polyurethane micro/nano composite. The experimental efforts were intended to compare the effectiveness of nano- and micro-particle sizes of Pb on gamma-ray attenuation properties. The radiation shielding characteristics of micro- and nano lead were experimentally determined for the 0.0595–1.41 MeV photon energy range. Moreover, the effectiveness of nano- and micro-particle sizes of Pb on polyurethane, the photon mass attenuation coefficients MAC, and the nuclear radiation shielding properties were simulated using the FLUKA code for the photon wide energy range of 0.0001–100 MeV.

## 2. Materials and Methods

### 2.1. Materials

The chemical components for rigid polyurethane foam were from DOW Chemical Company, Milan, Italy. Polyol raw material is a VORATHERM™CN 815, Methylene diphenyl diisocyanate (MDI) VORANATE™ M 600, Catalyst is a VORACOR™ CM, Catalyst is a VORACOR™ CM 639, The blowing agent is n-pentane, which was obtained from Climalife Company; toluene rectified was obtained from Piochem Chemical Company, Cairo, Egypt and lead metal powder with 100 mesh was obtained from ADWIC Chemical Lab, El-Gomhouria Chemical Company, Cairo, Egypt.

### 2.2. Characterization Techniques

#### 2.2.1. Transmission Electron Microscope (TEM)

To differentiate between micro- and nanosized Pb particles, we used a JEOL JEM-2100F transmission electron microscope, Tokyo, Japan, with a 200 kV acceleration voltage to conduct the TEM investigation.

#### 2.2.2. Scanning Electron Microscopy (SEM)

To examine and analyze the micro- and nanoparticle imaging characterization of solid objects. Scanning electron microscopy (SEM) (TM3030, Hitachi, Japan) was used to examine the cross-section shape and distribution of Pb within the samples. The samples were chopped into pieces. The accelerating voltage was set at 15 kV, and the magnification was 40×.

#### 2.2.3. Apparent Density Measurement

The density of the samples was determined in accordance with EN 1602 [26]. The samples were shaped like cylinders with a radius of 40 mm and a height of 80 mm. The volume of each sample was measured with a Vernier caliper, and the mass of the samples was weighed with an electronic balance (Weight Scale/Ls 1220 M, Precisa company, Dietikon, Switzerland) to calculate the apparent density.

#### 2.2.4. Compressive Stress Test

To determine the compressive stress, we use a universal testing machine (DVT FU/DLC Tensile/Compression Test Device DEVOTRANS Instrument, Istanbul, Turkey). The RPUF samples had a cylindrical shape with a diameter of 80 mm and a height of approximately 80 mm. According to ASTM D1621-1 [27], the samples were measured at a compression rate of 3 mm/min as specimens. Samples were compressed to 5% of the total thickness.

#### 2.2.5. Thermo-Gravimetric Analysis (TGA)

To evaluate the thermal stability of a material, a thermogravimetric analyzer (TG 209F3, NETZSCH, Selb, Germany) was used with dry nitrogen gas at a flow rate of 60 mL min^−1^ was used for the analysis. The samples’ relative mass loss was measured from 20 °C to 700 °C at a heating rate of 10 °C min^−1^.

#### 2.2.6. X-ray Diffraction Analysis (XRD) Analysis

The XRD characterization has been conducted to study the crystallinity of the nanostructured RPUF. A Rigaku (Model: D2 Phaser 2nd Gen—Bruker, Billerica, MA, USA) X-ray diffractometer with a Cu-K (0.154 nm) monochromatic radiation and optimal operating parameters of 30 mA and 30 kV was used to analyze the material. All analyses were performed at room temperature, scanning from 200 to 900 in 2 s. The scanning rate was 1.80 min^−1^, with a data gathering step size of 0.050. The powder was uniformly distributed on a double-sided sticky tape before being applied to the sample holder. After that, the holder was placed in the chamber to measure the angular distribution of X-ray diffraction (XRD) peaks. Using the Jade program, the obtained patterns were matched with a corresponding standard file.

#### 2.2.7. Fourier-Transform Infrared Spectroscopy (FTIR) Analysis

Fourier-transform infrared (FTIR) spectra were utilized to determine organic materials. The functionalized Pb was recorded using a Shimadzu dxp 400 (Kyoto, Japan), and the FT-IR spectrum was recorded in the mid-range between 4000 and 400 cm^−1^.

#### 2.2.8. TGA Analysis

TGA determines the quantity and the frequency of the weight variation of the samples against temperature and time in a controlled atmosphere (e.g., The purging of nitrogen gas). TGA can be used primarily to investigate the thermal stability (the strength of the material at a given temperature), and oxidative stabilities (the oxygen absorption rate of the material).

#### 2.2.9. Gamma-ray Acquisition Setup

The investigation of shielding properties of Pb MPs, and PbNPs polyurethane composites were performed by using a calibrated p-type Hyper Pure Germanium (HPGe) from Canberra company (Atlanta, GA, USA), its resolution of 7.5% at the 662 keV. A set of standard point sources (Am-241, Ba-133, Cs-137, Eu-152, and Co-60) have energy range from 0.060 keV to 1.408 Mev, Table 1 Practically to obtain a parallel beam of photons plus to minimize the coincidence summing effects [28] and the dead time to be below 1%the source was placed at a distance of 50.86 cm from the detector surface. The investigated sample was placed very close to the detector as shown in Figure 1.

The obtained spectra were analyzed by Genie 2000 software (Canberra company (Atlanta, Georgia, United States) to give the count rate at a particular energy. The signals from the detector were collected through a measuring time with a statistical error of less than 1%.

#### 2.2.10. Theoretical Background

The Radiation Shielding Parameters of interest are as follows:The linear attenuation coefficient (LAC) is the first step to evaluate the material capability for shielding, using Beer-Lambert’s law Equation (1) [29].
(1)$$\mathrm{LAC}=\mu =\frac{1}{t}\ln\left(\frac{I_{0}}{I}\right)$$

2.The mass attenuation coefficient (MAC) was calculated by dividing the LAC for a given material by its density (ρ). The MAC of a composition can also be calculated using Equation (2) [25]:(2)$$\mathrm{MAC}=\sum_{i}w_{i}{\left(\mathrm{MAC}\right)}_{i}$$
where (w_i_) is the mass-fraction of the ith constituent element in the sample. Using the XCOM program [30,31,32], the MAC theoretically was calculated and compared with the experimental values.

Using the values of MAC, the other shielding parameters were calculated such as the half-value layer (HVL), tenth-value layer (TVL), the mean free path (MFP) and the effective atomic number Zeff were calculated using Equations (3)–(6).


3.The average distance between two consecutive photon collisions is known as the mean free path (MFP), and it is determined using Equation (3) [33,34,35,36].


MFP = 1/µ(3)



4.The half-value layer (HVL) was calculated using Equation (4) [37,38].


HVL = ln2/µ(4)



5.Tenth value layer (TVL) was calculated using Equation (5) [39,40,41].


TVL = ln10/µ(5)


6.The effective atomic number Zeff is a crucial radiation interaction metric used to describe how radiation attenuation in composite materials varies depending on the specimen’s composition and photon energy. Using Equation (6) given below can be used to calculate the Z_eff_ values [42](6)$$Z_{\mathrm{eff}}=\frac{\sum_{i}\mathrm{fi}\mathrm{Ai}{\left(\mathrm{MAC}\right)}_{i}}{\sum_{i}\frac{\mathrm{Ai}}{\mathrm{Zi}}{\left(\mathrm{MAC}\right)}_{i}}$$
where fi, Ai, and Zi are weight fraction, atomic weight, and the atomic number of each constituent element in each specimen, respectively.

#### 2.2.11. FLUKA Simulations

Moreover, to confirm the validity of our results we used FLUKA Simulations beside the XCOM online software (NIST Standard Reference Database 8, National Institute of Standards and Technology: Gaithersburg, MD, USA). FLUKA is a general Monte Carlo (MC) Simulation platform code. It has many requests, for the interaction and transport of photons. It uses the best mathematical models and an accurate, microscopic approach [43,44]. FLUKA has a wide range of presentations, design shielding, radiation defense, dosimetry, and radiation detector simulation. The newest version of FLUKA, available on 13 January 2023 can be demanded and downloaded from the FLUKA site [45,46]. FLUKA needs an auxiliary code-named Flair. Flair is a user-friendly interface for FLUKA code to simplify the entrance of FLUKA input data, organizing FLUKA and visualization of the output tables and figures. It is completely based on Python mode. The latest flair-2.3-0 C code was published on 24 March 2023 and can be downloaded from the flair location site. In this activity, simulation was performed using the following FLUKA and flair factors:The beam profile is assumed to be rectangular.Abeam focused on the positive z-direction.GEOBEGIN card: combinational geometry is used in free arrangement.For BLKBODY and Void, a sphere is well-defined with R = 100,000 and R = 10,000 individually.Right Circular Cylinder code RCC declared for maple-target with HZ = 0.1 and R = 5.0.The MGDRAW.F all-purpose event subroutine interface was started by the USERDUMP card.The VOID district: ASSIGNMA VACUUM VOID.EMFCUT construction in a material is set equal to the bottom transport cut-offs in the requested area with the familiarized material.The introduced samples have been described by applying the constructed in MATERIAL and COMPOUND cards using C, H, N, O and Pb elements. All elements are pre-defined in the FLUKA default library.Simulation was conducted for 10^6^ primary particles and the code was run for five cycles. The results were directed to the output binary files of USRBIN and USRBDX. The average and mean values were calculated using the FLAIR RUN mode. Figure 2 shows the used simulated geometry.

### 2.3. Preparation of Nano Pb Using Ball Mill Machine

Milling was conducted by Photon Ball Mill Model PH-BML 912, Figure 3, Ball milling Jar Capacity: 250 (mL), rated rotating speed: revolution (big tray) 300 turns/min ± 10%; rotation (ball milling Jar): 600 rotations/min ± 10%.

Pure Lead powders of Pb (5 µm) converted by using Photon-planetary ball mill PM 600, Zirconia Milling Balls (10 mL–19 mL), ball to powder ratio 1:1, Toluene Rectified to prevent collection of particles, at 600 rpm after 5 h of milling, The average final size after grinding was 3 nm.

### 2.4. Composite Preparation

Rigid Polyurethane Foams (RPUF) are produced using the following steps (Figure 4):The Polyol raw material was mixed with a reactive blowing catalyst and trimerization catalyst, and mixed in a cartoon cup for 30 s with a mechanical stirrer at (2000 rpm) until uniform dispersions were obtained.Pb powder either lead microparticles (PbMPs) or lead nanoparticles (PbNPs) were mixed with the previous mixture for 30 s.Then the physical blowing agent was added to the mixture and mixed for a few seconds.Finally, isocyanate was added, and mixed for a few seconds.

The mixture was rapidly transferred into an open mold for foam rising. The RPUF was raised in the vertical direction, and then cured at room temperature for 24 h, samples were cut according to the standards of the different tests performed. The composition of polyurethane foams is given in Table 2.

## 3. Results and Discussions

### 3.1. Nano-Lead Preparation

Figure 5c shows the structure of the Pb microparticles, while Figure 5a shows the sample created by the ball mill machine. The particle size data were obtained from the counting of more than 300 particles. Particle size and size distribution data can be represented graphically as shown in Figure 5b, which summarizes the particle size data extracted from the TEM images. Based on the particle size data, the particle size distribution was plotted. A typical way to present particle size and its distribution is in the form of a number-frequency histogram. A histogram is a bar graph that illustrates the frequency of occurrence versus the size range, which shows that the average final size after grinding was 3 nm.

### 3.2. Effect of Lead on Cell Morphology and Density

Figure 6a shows the pure polyurethane foam compared with the polyurethane composite with 10%wt fillers. It was shown that the polymer doped by 10% either PbNPs or PbMPs showed a slight change in volume compared to the pure sample volume while using 50% wt PbMPs and 50% PbNPs showed a serious change in volume compared with the pure sample volume, as shown in Figure 6b. Also, in the case of the polyurethane composite with Nano, it was slightly bigger than that doped with micro fillers. This may be caused due to the high surface area of nanoparticles compared with microparticles. This advantage of nanoparticles helps to ensure homogenous distribution through polyurethane foam porosity during preparation and prevents the possibility of aggregation when using microparticle size.

Figure 7a shows the cell structure of a pure sample of polyurethane. The cell structures of polyurethane foam with the addition of lead nanoparticles and bulk sizes were analyzed using SEM as shown in Figure 7b–e. The use of lead provided the polyurethane with an open structure. The cells exhibited a diameter ranging between 170 and 76 µm. The microstructure of the samples with the weight fraction of fillers used shows that the cell distribution becomes more random and nonuniform compared to that in unreinforced polyurethane foam. Generally, the addition of Pb material caused a decrease in foam aperture size with a great range, which is apparent in 50% Pb, especially for micro size, as shown in Figure 7e. The porosity was high, but the cells lacked good evenness for 50% PbNPs, 10% PbMPs, and 50% PbMPs in ascending order, respectively. Compared to 50% PbNPs, other samples caused smaller cells, and the cells were compact and uniform. The difference was attributed to the surface tension and surfactant properties of the Pb. Moreover, with the addition of 10% PbNPs, the cell size pore decreased slightly, and the apparent density of the mixture correspondingly increased slightly from 34.08 to 35.64 kg/m^3^, as shown in Figure 8. Increasing the content of Pb adversely affected the cell uniformity of the mixture, except in cases of lower concentrations of PbMPs and PbNPs, where the cell size and porosity increased. The lower cell uniformity was attributed to the difference between particle size and percentage of lead, which in turn changed the cell structure of the mixture sample. However, this increase was significant because the large particle size decelerated the nucleation reaction of the foam, increased the crosslinking degree and prepolymer density, and decreased the amount of released gas. Figure 8 shows a high difference in apparent density between PbNPs and PbMPs at different percentage ratios.

### 3.3. Effect of Lead on Compressive Stress

Compressive Stress is an important parameter that impacts the performance characteristics of the foam, and the change in its value is presented in Figure 9a–c. The compressive strength of all materials tested was measured in a direction perpendicular to the direction of foam rise. In the current study, rigid polyurethane foam was weakened with different contents of Pb nano and microparticles (10 and 50 wt%). A compression test was performed in the range of 5% of the total sample thickness. The results showed that the addition of Pb nanoparticles impaired the compression properties of the foam. The largest increase in compressive strength was observed for the samples that contained 10 wt% fillers, and their load was about 960 kPa in the perpendicular direction. As we increase the percentage of Pb fillers from 10 wt% to 50 wt%, there is a decrease in compressive strength compared with pure foam; the compressive strength decreases from 990 to 140 kPa in the perpendicular direction. Nonetheless, similar to their Pb micro filler counterparts, the compressive strength of PU-modified foams is higher compared to that of Pb nanofiller foams. The microstructure of the samples with the weight fraction of fillers used shows that the cell distribution becomes more random and nonuniform compared to that in unreinforced polyurethane foam. The loss of uniform microstructure may result in scattered areas that are weak in compression. Once these areas begin to fail in compression, the whole cross-section progressively buckles under the compressive load.

### 3.4. Thermo-Gravimetric Analysis (TGA)

A thermogravimetric analysis for RPUF was studied with and without the addition of Pb filler. The heating rate was set to 10 °C/min and measured in a nitrogen atmosphere. The samples were heated to 500 °C. There was a two-stage degradation. Urethane linkages were broken down at lower temperatures (first stage) with the evolution of isocyanate and alcohol. Urea linkages and polyol bonds were cleaved when enough energy was provided at high temperature (Second stage) forming complex products. When Pb fillers were added to polyurethane foam with different content ratios, the weight loss changed as the temperature increased. The results are shown in Figure 10, and Table 3 shows the changes in the decomposition curves. The initial decomposition temperature (IDT) for RPUF was 240 °C when the weight loss was 8%. After Pb filler particles were added to RPUF, and as the concentration of Pb filler increased, IDT increased considerably to 300 °C (PbMPs 50% composite) when the weight loss of the hybrid material was 8%. As the concentration of Pb filler particles increased, the remaining mass increased after completely degrading the substance, including residual carbon, particularly at 500 °C. Figure 10 shows the large differences in weight retention between different samples, revealing that Pb filler increased the thermal stability of RPUF. The derivative thermogravimetric curves are displayed in Figure 10. As shown, the control group exhibited the lowest carbon residue amount. Moreover, RPUF containing 50% nano- and micro-pb showed the greatest amount of carbon residue. As shown in Table 3, this indicates that the addition of Pb has a significant effect on the degradation of RPUF and leads to an increase the thermal stability. In the polyurethane foam without the addition of Pb, the surface of the carbon residue was continuous. RPUF with 50% Pb shows the highest thermal stability due to its high initial decomposition temperature and has the lowest rate of degradation. The above proved that the addition of Pb filler nanoparticles effectively improved the thermal stability of hybrid materials.

### 3.5. XRD Analysis

Polyurethane foam with and without fillers was analyzed using X-ray diffraction (XRD), a non-destructive analytical method for identifying the atomic and molecular structure of materials. For the pure polyurethane foam, the XRD pattern in Figure 11 shows a wide diffraction peak without any discernible sharp peaks, indicating an amorphous molecular structure. This microstructural trait fits with the PUE material’s translucent property. The wide diffraction peak in this material shows up at roughly 20 degrees, which is in line with what has been observed in other polyurethane materials. It was shown that the broad peak that indicates the presence of polyurethane foam slightly decreased when 10% Pb was added, while the peak passed out of sight at 50% Pb was added for both sizes. As can be seen, several remarkable peaks at 29.09, 32.30, 38.09, 49.53, and 63.63 °C are attributed to the Pb substrate [47]. The resultant peaks of 50% micro Pb are sharper and have a width at half maxima lower than that of 50% Nano Pb.

### 3.6. FTIR Analysis

Polyurethane rigid foam is a kind of high polymer with a basic repetitive unit urethane bond (−NHCOO−) and can be produced from the polymerization of isocyanates and polyols. The FTIR spectroscopy result of rigid PU foam in air at room temperature (RT) is shown in Figure 12. (Spectral range from (4000 to 500 cm^−1^). For isocyanates, symmetrical and asymmetric segments Figure 12. stretching vibrations of N−H correspond to the broad absorption bands near 3361.17 cm^−1^ and 3289.80 cm^−1^, while the medium-strong peak at 1508.98 cm^−1^ corresponds to the in-plane bending vibration of N−H. The sharp absorption peaks at around 1706.15 cm^−1^ and 1267.66 cm^−1^ are typical for the stretching vibration of esters C=O and asymmetric stretching vibration of C−O (from N−CO−O), respectively. The weak peak near 815.31 cm^−1^ is due to the N−CO−O symmetric stretching vibration. The 1597.49 cm^−1^ absorption peak is caused by the vibration of C=C in the benzene ring and the several weak peaks near 760–815.31 cm^−1^ belong to the out-of-plane bending vibration of C−H in the multi-substituted benzene ring. For soft segment polyether polyols, 2904.35 cm^—1^ is caused by C−H stretching vibration. The –CH_3_ bond shows the symmetric bending vibration at 1380 cm^−1^. Bond C−O−C stretch results in the broad and strong peak at 1071.71 cm^−1^. This results in good agreement with the literature [48].

### 3.7. Fluka Results

This work aimed to investigate the ionizing radiation shielding features of Lead/Polyurethane Micro/Nano Composite. The mass attenuation coefficients (μ/ρ) were generated by XCOM software over an extended photon energy range from 0.0595 MeV up to 1.4080 MeV. The obtained data were verified via the calculated data from FLUKA Monte Carlo simulation code The obtained values of μ/ρ were used to evaluate several important shielding parameters such as half value layer (HVL), mean free path (MFP), effective atomic number (Zeff). The calculated MAC of lead free and 10–50% samples coded S1–S6, respectively, at different incident energies is plotted in Figure 13. The FLUKA simulations can predict more results for the energy range 0.0001–100 MeV. The benefit of using FLUKA is that the energy interval can be distributed to any portion to give detailed results that are more precise. It can predict all the peaks (tooth) inside the examiner range. In this calculation, 5000 portion intervals are preferred for the range from 0.1 keV instance photon energy higher to 100 MeV. This feature shows that the region ranging from 0.1 keV up to 100 MeV has three tooth peaks with photon energies 0.0025, 0.014 and 0.1 MeV. All samples’ results overlapped near 1–2.8 MeV photon energy. It is clear that the sample of Pb free and 10% Pb has the lowest MCA value, while the 50% Pb sample has the highest one. The XCOM calculations confirm the FLUKA results as shown in Table 4.

Figure 14 shows the dependence of the LAC (μ) with the incident photon energy (MeV). Its value ranges from 1500–~0.01 (cm^−1^) for the energy range 1 × 10^−3^ up to 100 MeV. The obtained curves show the same trends as MAC results. Moreover, the mean free path values MFP of the proposed samples are given. It takes values 0.07–150 m for the energy range of 0.0001 up to 100 MeV. The HVL (T_1/2_) and the TVL (T_1/10_), Figure 15, of the S1–S6 samples almost take the values 4.84 × 10^−4^–6.6 × 10^−1^ and 1.2 × 10^−3^–5.78 × 10^2^, respectively. The effective atomic number, Z_eff_ of the proposed samples are given in Figure 16. It is clear that samples have a peak value near 0.1 MeV which does not appear for the Pb-free sample. Z_eff_ takes values (6.1–434 for lead free), (40.71–5.75 for 10% Pb sample) up to (73.12–15.38 for 50% Pb sample).

### 3.8. Experimental Results of Attenuations Coefficients

Based on the test results of the experiments, the linear attenuation coefficient (LAC), was calculated for all investigated composite samples, using the sample densities to calculate the mass attenuation coefficient (MAC), these results were tabulated in Table 4 and Table 5. To confirm the validity of these experimental results. We applied XCOM and FLUKA programs to Pb-free samples Table 4. Where the mass attenuation coefficient (MAC) was simulated by FLUKA and XCOM online software and compared with the experimental MAC values. The deviation, or difference percentage, between the theoretical and experimental results, was given by the following Equation (7).
(7)$$\mathrm{Deviation}\;\left(\%\right)=\frac{{\mathrm{MAC}}_{\mathrm{Th}}-{\mathrm{MAC}}_{\exp}}{{\mathrm{MAC}}_{\exp}}\times 100$$

For the XCOM and the FLUKA, the maximum absolute relative deviation from the experimental values was 6.61% and 7.82 at 0.1218 MeV, while the minimum deviation was 0.32% and 0.46 at 0.9641 MeV, which indicated a suitable agreement between the experimental and XCOM and FLUKA results, and this is a prediction of the experimental accuracy.

The slight nonconformity between the theoretical and experimental results has been the focus and interest of many researchers [50]. It can be due to the effect of the collimator size, hole, and absorber thickness. For gamma rays, the accurate measurements of MAC (cm^2^ g^−1^) ideally need perfect narrow beam irradiation geometry. However, usually, the practical geometries used for the experimental investigations deviate from perfect narrowness, thereby the multiple scattered photons cause systematic differences in the measured values of MAC. Good agreement between theoretical and measured values of MAC was observed for all absorbers with a thickness less than or equal to 0.5–1 mean free paths, thus considered as optimum thickness for low-Z materials in the selected energy range. Sometimes it is very difficult to prepare samples with these specifications in order to satisfy these measurements. This leads us to use the available samples in sizes and volumes that we can suitably manufacture.

On the other hand, FLUKA uses different libraries: BROND-3.1, ENDF-VIII0, JEFF-3.3, and JENDEL-4.0. The photon interaction libraries contain records to define the interaction of photons with the elements Z = 1–100 over the energy range up to 100 MeV. These libraries have been designed to meet the needs of users to model the interaction and transport of primary photons. It should be mentioned that using the default JENDEL library through the simulation process may be one of the reasons for such deviations, especially in the low-Z mode.

In Table 5 and Figure 17 and Figure 18, the experimental results were discussed at different energies, from 0.0595 MeV up to 1.4080 MeV, and it was clear that the highest MAC corresponded to low energy and the values decreased gradually until they reached the highest energy. For low energy photons < 150 keV where the predominant interactions are due to photon electric effect, due to the presence of Pb one will notice a remarkable change in the values of LAC in the energy region (88 to 120 keV) because Pb has K-edge at 88 keV energy. While in the energy region greater the 150Kev the Compton scattering is the predominant interaction. For greater energies, the effect of Pb on the photon attenuation properties of polyurethane decreases due to the increase in the penetration of energetic photons and the increase in mean free path.

Figure 17 displays the result of the LAC (cm^−1^) of the lead-free sample, which was evaluated by interring the density factor.

The experimental MAC results for other composites (containing micro and nano Pb concentration) were determined and the micro and nanocomposites were compared together as shown in Table 5. The relative difference between these results is given by the following Equation (8).:(8)$$\mathrm{Dev}1\;\left(\%\right)=\frac{{\mathrm{MAC}}_{\mathrm{nano}}-{\mathrm{MAC}}_{\mathrm{micro}}}{{\mathrm{MAC}}_{\mathrm{micro}}}\;\times 100$$

From the MAC values and density, the LAC results were estimated for both micro and compositions as shown in Table 4 and Table 5 and the relative deviation between these results is given by the following Equation (9).:(9)$$\mathrm{Dev}2\;\left(\%\right)=\frac{{\mathrm{lAC}}_{\mathrm{nano}}-{\mathrm{lAC}}_{\mathrm{micro}}}{{\mathrm{lAC}}_{\mathrm{micro}}}\;\times 100$$

It is clear that the MAC values for nano are greater than those of composites at all energies with all Pb percentages (10, 20, 30, 40, and 50%), as shown in Table 5 and Figure 18. Where the relative deviation (Dev1) was always positive, these results were due to the greater distribution of Pb nanoparticles in the matrix rather than the Pb microparticles, since the better distribution enhanced the attenuation ability of the composite. In other words, the LAC values are affected and depend on the density of the composite rather than any other factor, so we can see the results of 10% and 20%. LAC values have the same characteristics as MAC values due to the small difference in density between these two composites, while in 30%, 40%, and 50% composites, the difference in density between the two composites is clear, with the density of micro-composites being greater than that of nano-composites. For this reason, we can see that the LAC values for nano are smaller than those for micro composites at all energies with Pb percentages (30, 40, and 50%), as shown in Table 5 and Figure 19. The attenuation results of the Lead/Polyurethane micro and nano Composites.

The half-value layer (HVL) and mean free path (MFP) of the pure, 10% and 50% lead size percentages were calculated and tabulated in Table 6. The results indicated that the HVL and MFP values increased with increasing photon energy as shown in Table 6 and Figure 20 and Figure 21. The values of HVL and MFP decrease with increasing lead percentages in the matrix polymer composites; for example, the value of HVL at 50% microlead/polymer is the lowest value of the corresponding micro-composites at all discussed energies. In other words, the half-value layer and mean free path values of the composites impeded with nano-lead are lower than those of the micro-composites in all energy ranges.

Figure 22 shows the Zeff values of prepared polymer composites against photon energy. It is clear that the values of the effective atomic number (Zeff) decreased with increasing the photon energy until the energy of 0.122 MeV of the 152Eu γ-ray source affected by the k-edge of the lead element at ~0.088 MeV increased at this point and then decreased gradually with increasing the photon energy. The Zeff values also increased with increasing the Pb concentration inside the present polymer, where the Zeff of 50% was greater than 40%, etc. The maximum value was 48.43 of 50% lead/polymer at 0.0595 MeV, while the lowest Zeff value was 3.90 of pure polymer at 1.4081 MeV. The values of HVL and MFP decrease with increasing lead percentages in the matrix polymer composites.

## 4. Conclusions

A lead/polyurethane composite is made using the in-situ polymerization process successfully with different concentrations after the preparation of nano lead by a ball mill machine with specific conditions, which is supported by XRD results; 50% PbMPs and 50% PbNPs density and morphology tests showed a serious change in the volume of the pure sample and the pore size compared with the control sample. Generally, the addition of Pb material caused a decrease in foam aperture size with a great range, which increased the densities, especially for the micro size. The cell distribution becomes more random and nonuniform causing impaired the compression properties of the foam. TGA indicates that the addition of Pb increases the thermal stability. Results of FTIR generally showed no big difference in peak position, but the resultant peaks of 50% microparticles Pb were sharper and had a width at half maxima lower than that of 50% nanoparticles, which may be caused by the high surface area of nanoparticles compared with microparticles.

Shielding from gamma rays (or photons) is crucial for some applications. Polyurethane is used in construction. We can improve its shielding properties with some additives like Pb, which is all found in pure polyurethane and has the lowest shielding capability, by adding 10–50% PbMPs and PbNPs. The MAC values for nano-composites are greater than the micro composites at all energies with all Pb percentages (10, 20, 30, 40, and 50%); the LAC values for nanocomposites are smaller than the micro composites at all energies with Pb percentages of 30, 40, and 50% and the Zeff values also rise with increasing lead concentration inside the present polymer. Moreover, the nuclear radiation shielding capabilities were thoroughly examined for samples containing a Pb ratio of up to 50% and for lead-free samples. For this target, calculations for a wide energy range of 0.0001–100 MeV photon energy were achieved by the FLUKA simulation code. Based on the MAC values, other vital photon-shielding parameters, HVL, TVL, and Zeff, were also given and discussed.

The modified RPUF composite can be used in sandwich panel walls, used to fabricate mobile caravans, especially X-ray caravans that can be used in X-ray rooms, which is an important contributor to protecting employee health in X-ray rooms.

## Figures and Tables

**Figure 1 polymers-15-04416-f001:**
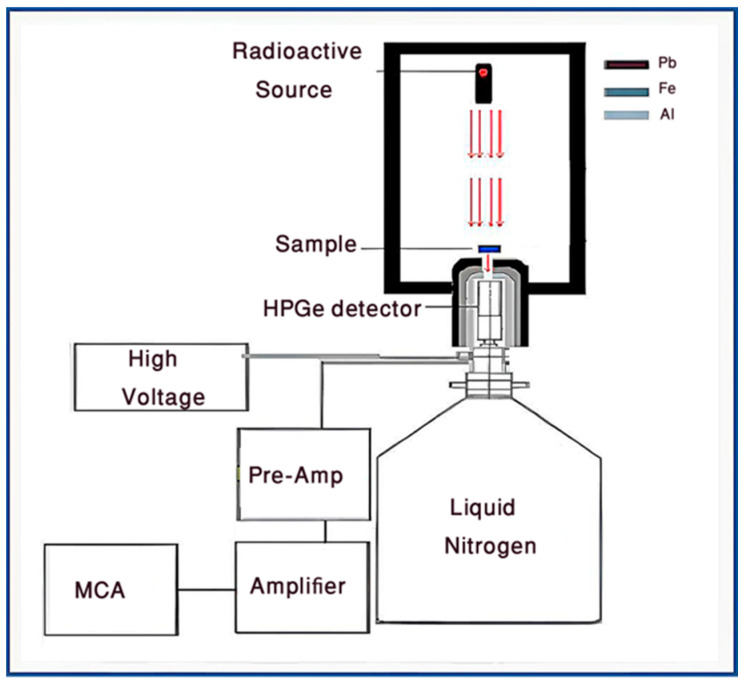
Experimental setup for a (HPGe) detector.

**Figure 2 polymers-15-04416-f002:**
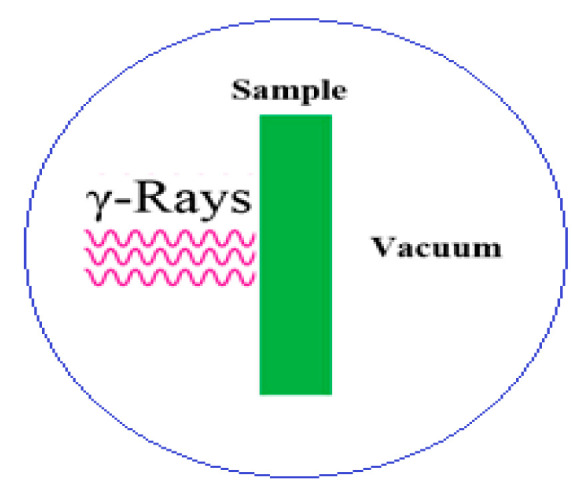
The simulation setup for FLUKA code. Samples were modelled as a cylinder, 10 cm diameter, with various thicknesses 1–5 mm.

**Figure 3 polymers-15-04416-f003:**
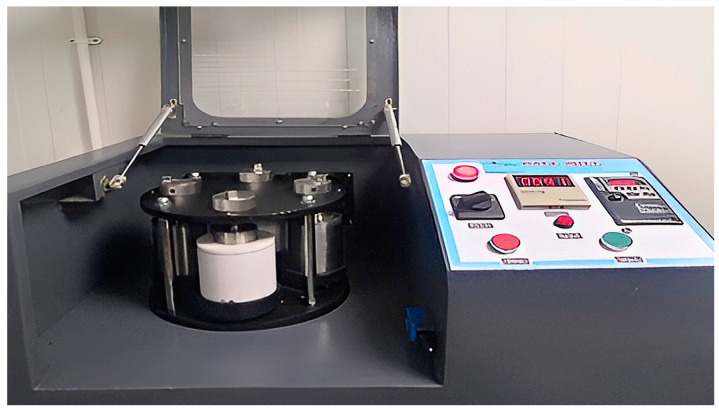
Ball Mill machine.

**Figure 4 polymers-15-04416-f004:**
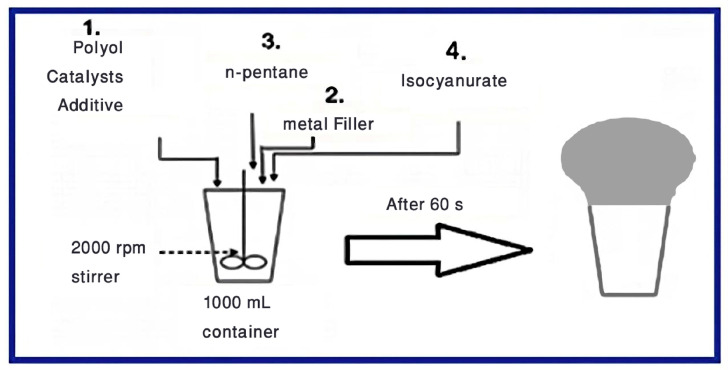
Polymer preparation scheme.

**Figure 5 polymers-15-04416-f005:**
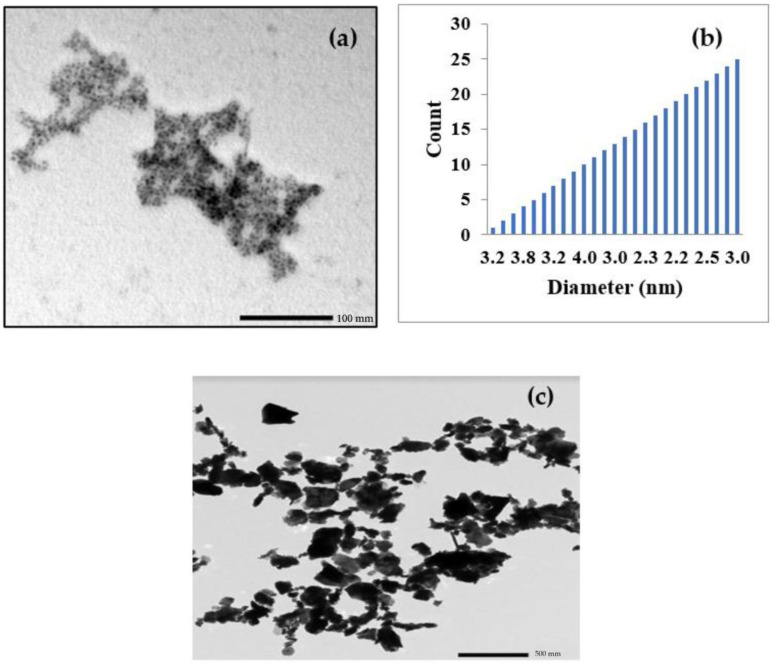
TEM images of (**a**,**b**) Pb NPs (**c**) Pb MPs.

**Figure 6 polymers-15-04416-f006:**
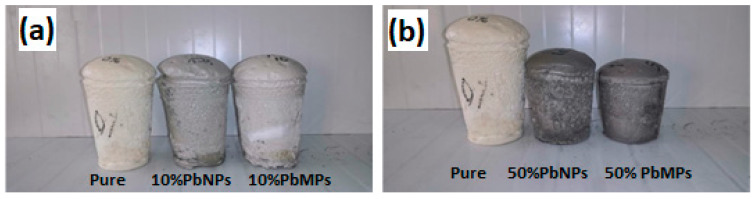
(**a**)Polyurethane composite foam 10% wt and (**b**) Polyurethane composite foam 50%.

**Figure 7 polymers-15-04416-f007:**
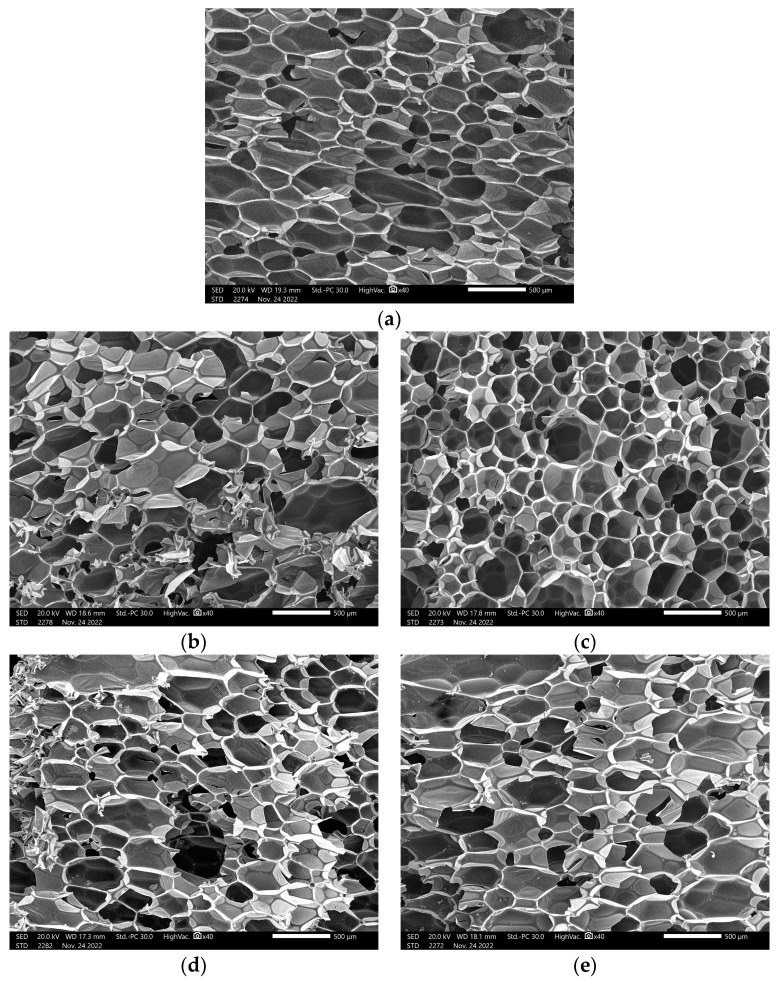
(**a**) SEM of pure polyurethane foam, (**b**) SEM of polyurethane foam +10% PbMPs Mixture, (**c**) SEM of polyurethane foam +10% PbNPs Mixture, (**d**) SEM of polyurethane foam +50% PbMPs Mixture, (**e**) SEM of polyurethane foam +50% PbNPs Mixture.

**Figure 8 polymers-15-04416-f008:**
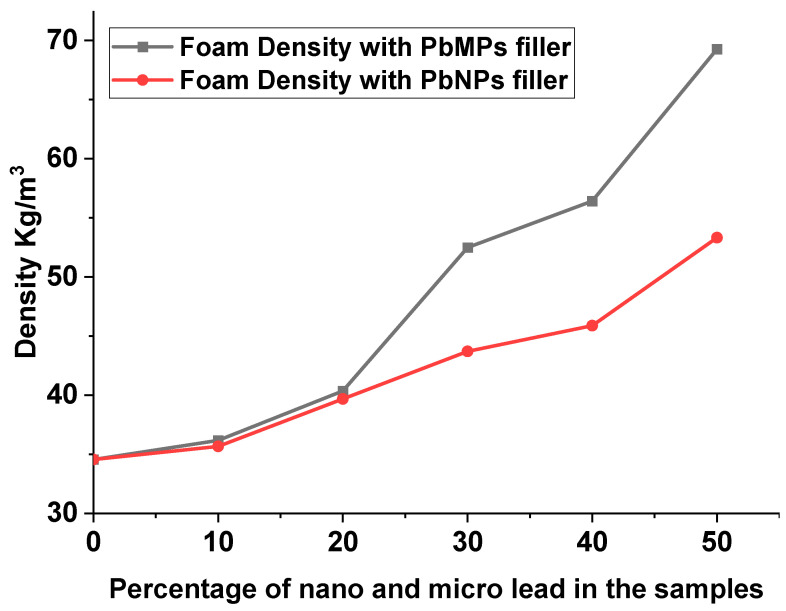
Foam Density comparison with different percentages of nano and micro lead filler.

**Figure 9 polymers-15-04416-f009:**
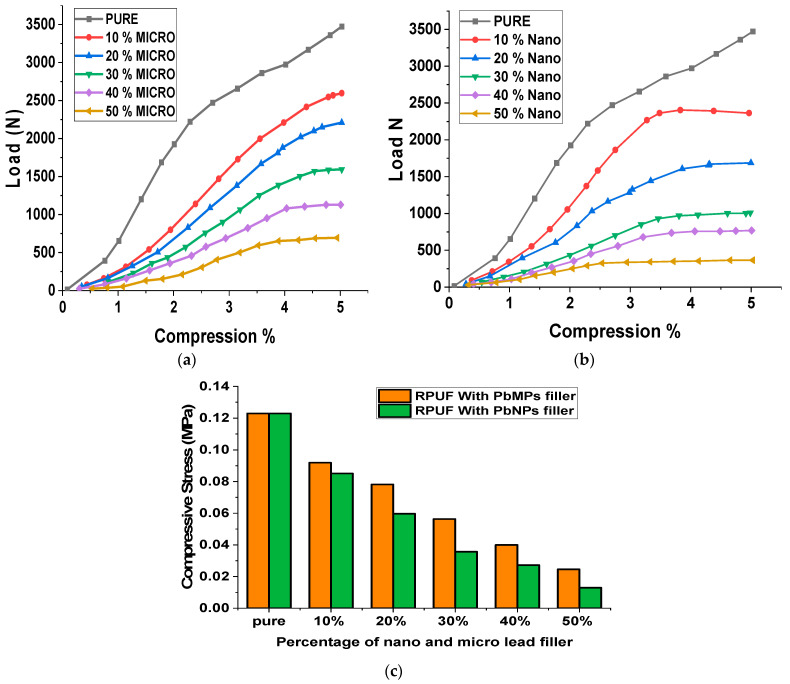
(**a**) Compressive Strength of RPUFs with Micro lead, (**b**) Compressive Strength of RPUFs with Nano lead, (**c**) Compressive Strength of RPUFs with different percentages of nano and micro lead filler.

**Figure 10 polymers-15-04416-f010:**
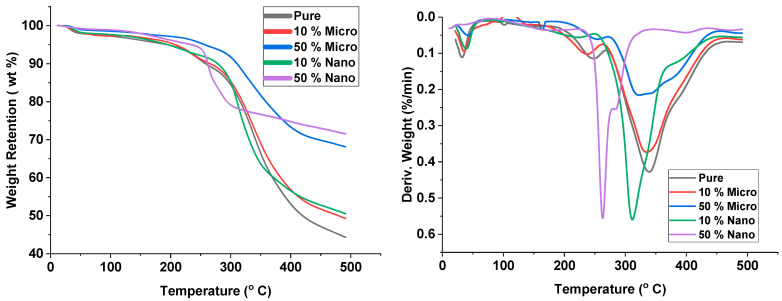
TGA curves of FPUF with and without lead addition.

**Figure 11 polymers-15-04416-f011:**
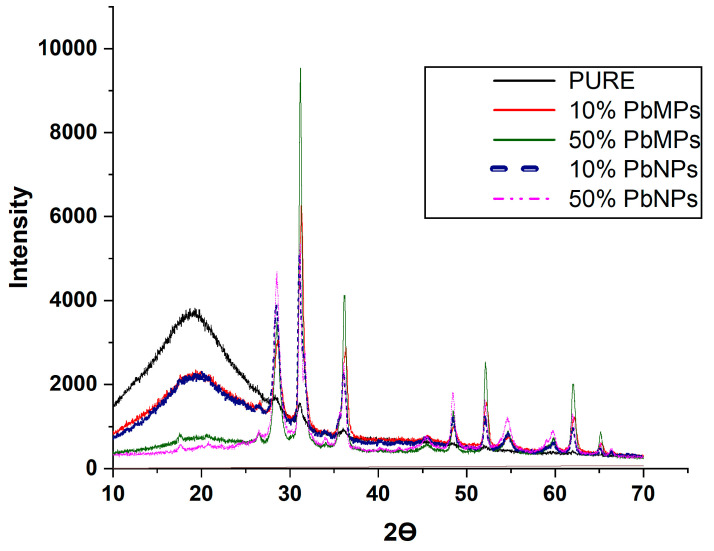
XRD-curves of RPUF with and without lead additions.

**Figure 12 polymers-15-04416-f012:**
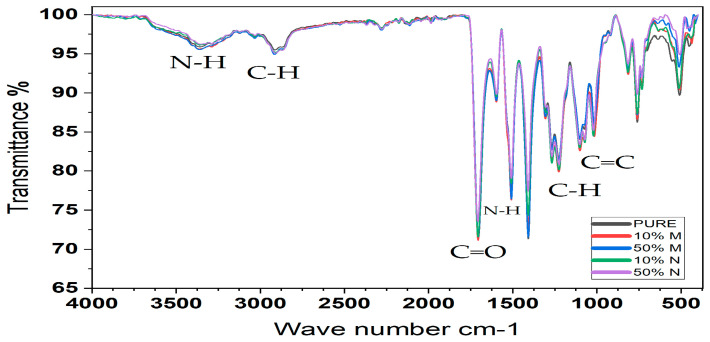
FTIR curves of FPUF with and without lead addition.

**Figure 13 polymers-15-04416-f013:**
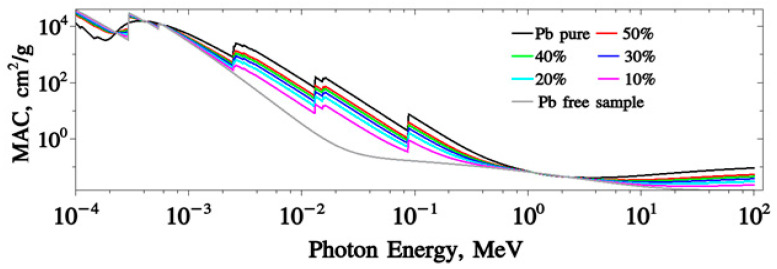
The (MAC) calculations for S1–S7 samples using FLUKA simulations.

**Figure 14 polymers-15-04416-f014:**
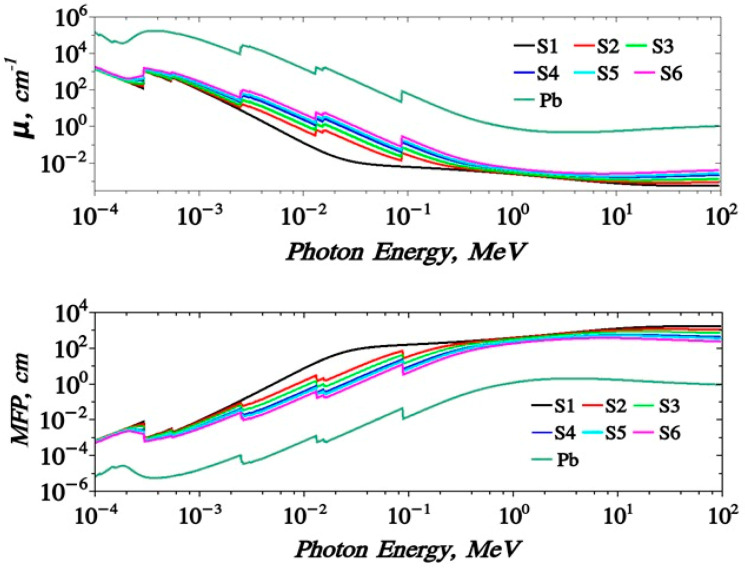
Variation of (LAC, μ) and (MFP) against photon energies for S1–S6 samples.

**Figure 15 polymers-15-04416-f015:**
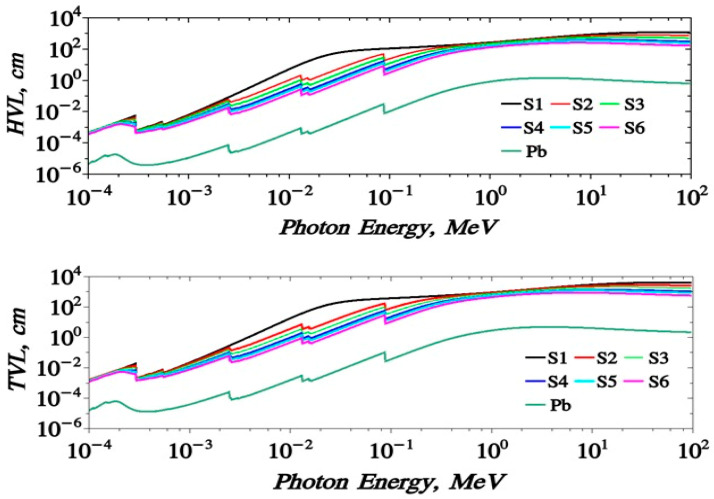
Variation of (HVL) and (TVL) against photon energies for S1–S7 samples.

**Figure 16 polymers-15-04416-f016:**
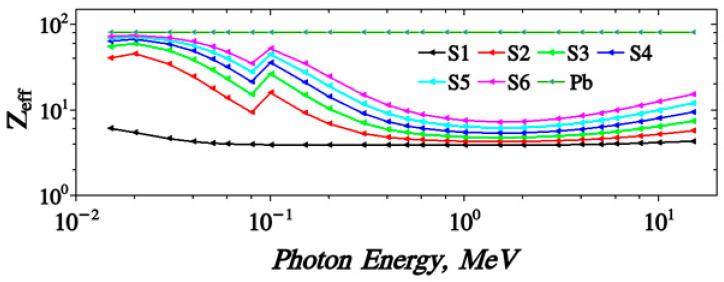
Variation of Z_eff_ against photon energy for S1–S6 samples.

**Figure 17 polymers-15-04416-f017:**
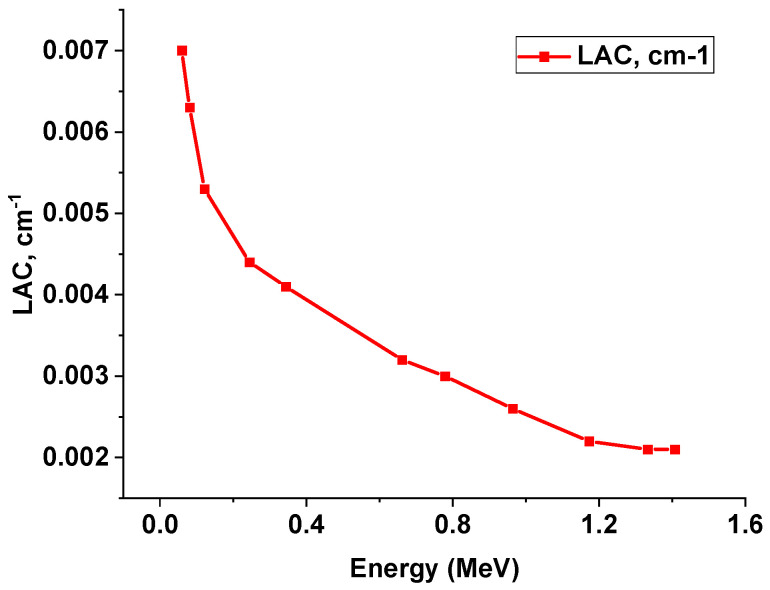
Experimental values of LAC (1/cm) of lead-free sample.

**Figure 18 polymers-15-04416-f018:**
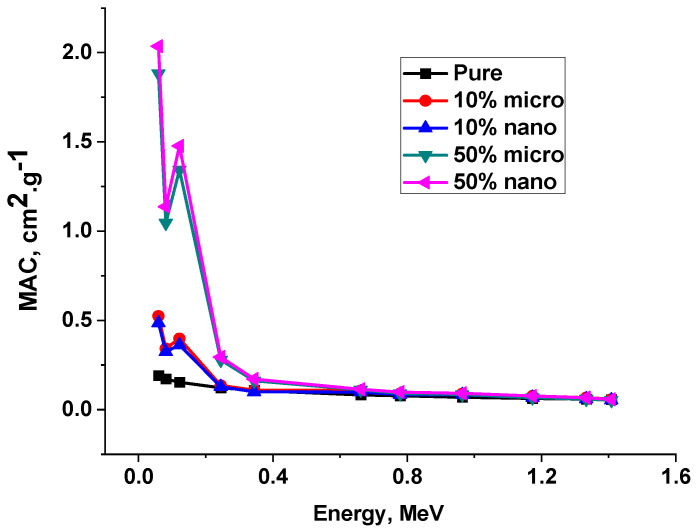
The MAC of pure and 10% and50% Lead/Polyurethane micro and nano Composites.

**Figure 19 polymers-15-04416-f019:**
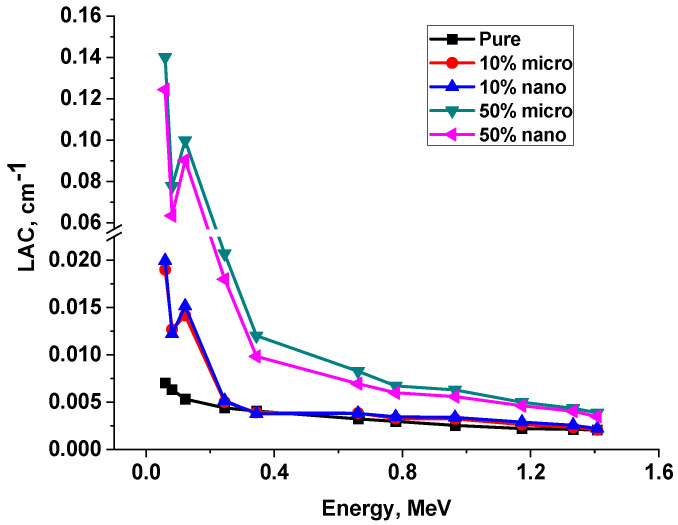
The LAC of pure and 10% and 50% Lead/Polyurethane micro and nano Composites.

**Figure 20 polymers-15-04416-f020:**
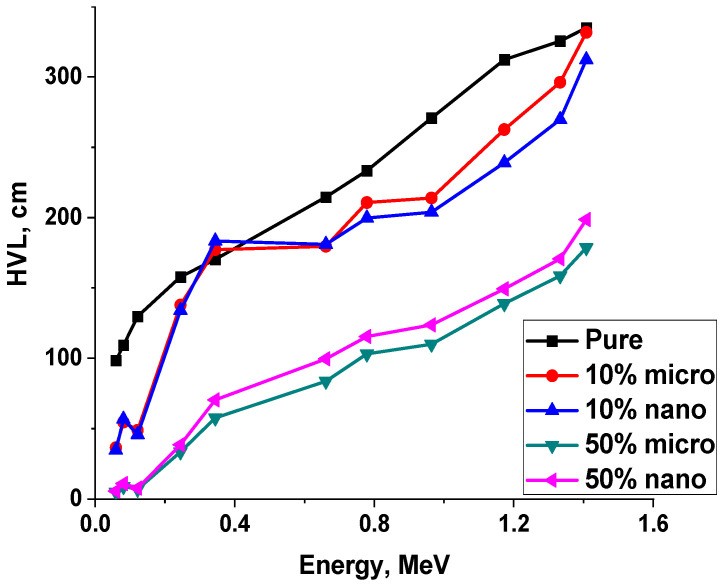
The HVL of pure and 10–50% Lead/Polyurethane micro and nano Composites.

**Figure 21 polymers-15-04416-f021:**
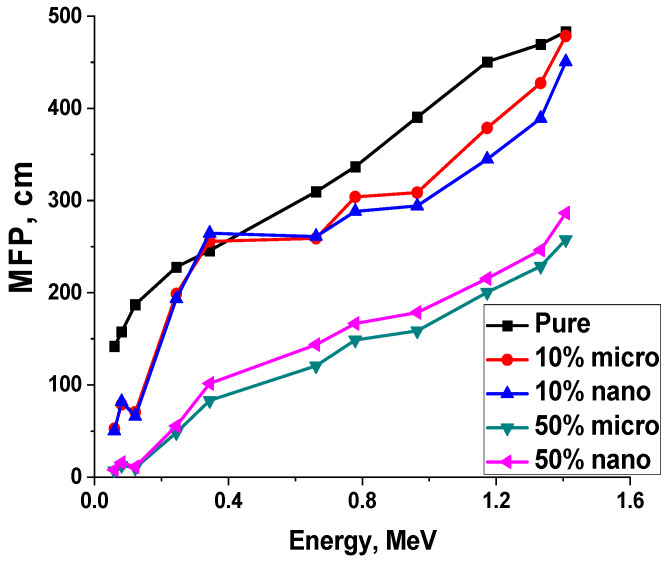
The MFP of pure and 10–50% Lead/Polyurethane micro and nano Composites.

**Figure 22 polymers-15-04416-f022:**
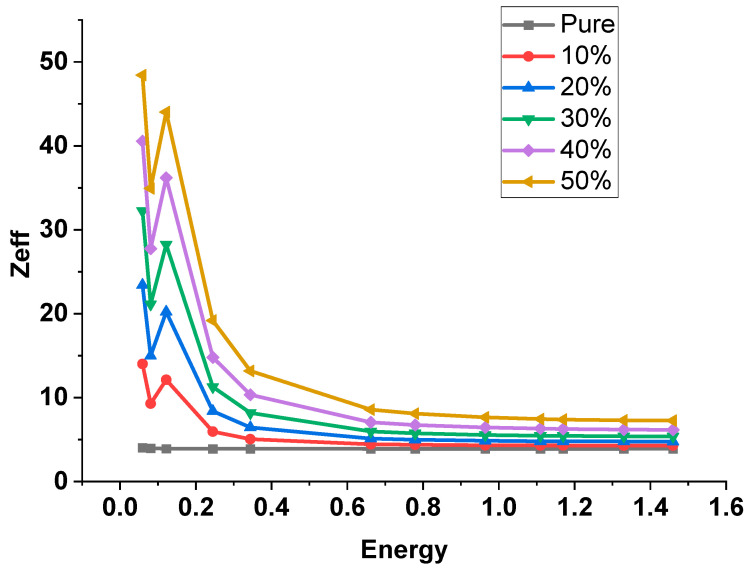
The Z_eff_ of Pb free and (10–50%) Lead/Polyurethane vs. photon energy.

**Table 1 polymers-15-04416-t001:** Photon energies, and half-life time for all radioactive sources for all radionuclides used in this work.

PTB-Nuclide	Energy (keV)	Half Life (Days)
241Am	59.52	157,861.05
133Ba	80.99	3847.91
152Eu	121.78	4943.29
	244.69	
	344.28	
	778.9	
	964.13	
	1408.01	
137Cs	661.66	11,004.98
60Co	1173.23	1925.31
	1332.5	

**Table 2 polymers-15-04416-t002:** Composition of polyurethane foams.

Component	PUF	PbMPs 10%	PbMPs 50%	PbNPs 10%	PbNPs 50%
Pb Micro	0.00	7.86	39.29	0.00	0.00
Pb Nano	0.00	0.00	0.00	7.86	39.29
Polyol	25.00	22.50	12.50	22.50	12.50
Isocyanate	48.75	43.88	24.38	43.88	24.38
Reactive blowing Catalyst	0.25	0.23	0.13	0.23	0.13
Trimerization Catalyst	0.58	0.52	0.29	0.52	0.29
N-Pentane	4.00	3.60	2.00	3.60	2.00

**Table 3 polymers-15-04416-t003:** Thermo-gravimetric analysis (TGA) data.

PRUFs	Decomposition Temperature °C (IDT)	Temperature °C		Weight Loss % at 490 °C
		Weight Loss 10%	Weight Loss 20%	
Pure	240	260	320	55.7
10% Nano Pb	260	290	315	49.44
50% Nano Pb	250	265	300	28.45
10% Micro Pb	240	280	330	50.75
50% Micro Pb	300	330	375	31.9

**Table 4 polymers-15-04416-t004:** Experimental and theoretical values of the MAC of a lead-free sample.

Sample Code	Energy * (MeV)	LAC (cm^−1^) EXP	MAC (cm^2^/g)			Dev. from XCOM	Dev. from FLUKA
			EXP.	XCOM	FLUKA		
Lead Free Sample density 0.0373 (g/cm^3^)	0.0595	0.0070	0.1888	0.1910	0.1952794	0.68	3.43
	0.0810	0.0063	0.1698	0.1717	0.174961	1.13	3.04
	0.1218	0.0053	0.1430	0.1525	0.1541774	6.61	7.82
	0.2447	0.0044	0.1177	0.1220	0.1226736	3.64	4.23
	0.6617	0.0032	0.0867	0.0820	0.0822526	−5.42	−5.13
	0.7789	0.003	0.0795	0.0775	0.0763563	−2.54	−3.95
	0.9641	0.0026	0.0686	0.0688	0.0689176	0.32	0.46
	1.1732	0.0022	0.0594	0.0624	0.0625992	5.18	5.39
	1.3325	0.0021	0.0571	0.0585	0.0585721	2.36	2.58
	1.4080	0.0021	0.0555	0.0568	0.0569979	2.32	2.70

* Photon energy taken from the IAEA live chart of the nuclides site [49].

**Table 5 polymers-15-04416-t005:** The attenuation results of the Lead/Polyurethane micro and nano Composites.

Sample Code		MAC, cm^2^·g^−1^			LAC, cm^−1^		
	Energy (MeV)	Micro Pb	Nano Pb	Dev 1	Micro Pb	Nano Pb	Dev 2
10%	0.0595	0.4854	0.524	7.95	0.019	0.02	4.92
	0.0810	0.3239	0.342	5.59	0.0127	0.013	2.63
	0.1218	0.362	0.3978	9.89	0.0142	0.0152	6.81
	0.2447	0.1285	0.1358	5.68	0.005	0.0052	2.72
	0.6617	0.0984	0.1066	8.33	0.0039	0.0041	5.29
	0.7789	0.084	0.0911	8.45	0.0033	0.0035	5.41
	0.9641	0.0826	0.0893	8.11	0.0032	0.0034	5.08
	1.1732	0.0675	0.0763	13.04	0.0026	0.0029	9.87
	1.3325	0.0598	0.0676	13.04	0.0023	0.0026	9.87
	1.4080	0.0533	0.0582	9.19	0.0021	0.0022	6.13
	The Density of Micro-Pb and Nano-Pb were (0.0392 and 0.0381 g·cm^−3^)						
20%	0.0595	0.8355	0.9012	7.86	0.0363	0.0387	6.38
	0.0810	0.5046	0.5497	8.94	0.022	0.0236	7.44
	0.1218	0.6076	0.6671	9.79	0.0264	0.0286	8.28
	0.2447	0.166	0.1753	5.6	0.0072	0.0075	4.15
	0.6617	0.1017	0.1038	2.06	0.0044	0.0045	0.66
	0.7789	0.0856	0.0928	8.41	0.0037	0.004	6.92
	0.9641	0.0832	0.0898	7.93	0.0036	0.0039	6.44
	1.1732	0.0674	0.0761	12.91	0.0029	0.0033	11.35
	1.3325	0.0596	0.0672	12.75	0.0026	0.0029	11.2
	1.4080	0.053	0.0579	9.25	0.0023	0.0025	7.74
	The Density of Micro-Pb and Nano-Pb were (0.0435 and 0.0429 g·cm^−3^)						
30%	0.0595	1.1847	1.2781	7.88	0.0679	0.0619	−8.87
	0.0810	0.6848	0.7482	9.26	0.0392	0.0362	−7.71
	0.1218	0.8525	0.9363	9.83	0.0488	0.0453	−7.23
	0.2447	0.2033	0.2147	5.61	0.0116	0.0104	−10.8
	0.6617	0.1049	0.1071	2.1	0.006	0.0052	−13.76
	0.7789	0.0871	0.0945	8.5	0.005	0.0046	−8.36
	0.9641	0.0836	0.0903	8.01	0.0048	0.0044	−8.76
	1.1732	0.0673	0.076	12.93	0.0039	0.0037	−4.61
	1.3325	0.0593	0.0669	12.82	0.0034	0.0032	−4.71
	1.4080	0.0527	0.0576	9.3	0.003	0.0028	−7.68
	The Density of Micro-Pb and Nano-Pb were (0.0573 and 0.0484 g·cm^−3^)						
40%	0.0595	1.5342	1.6566	7.98	0.0898	0.0803	−10.48
	0.0810	0.8652	0.8996	3.98	0.0506	0.0436	−13.8
	0.1218	1.0978	1.2066	9.91	0.0642	0.0585	−8.88
	0.2447	0.2407	0.2545	5.73	0.0141	0.0123	−12.34
	0.6617	0.1081	0.1105	2.22	0.0063	0.0054	−15.25
	0.7789	0.0887	0.0963	8.57	0.0052	0.0047	−9.99
	0.9641	0.0842	0.091	8.08	0.0049	0.0044	−10.4
	1.1732	0.0672	0.0759	12.95	0.0039	0.0037	−6.36
	1.3325	0.059	0.0667	13.05	0.0035	0.0032	−6.27
	1.4080	0.0524	0.0573	9.35	0.0031	0.0028	−9.34
	The Density of Micro-Pb and Nano-Pb were (0.0585 and 0.0485 g·cm^−3^)						
50%	0.0595	1.8829	2.0352	8.09	0.1401	0.1246	−11.09
	0.0810	1.0452	1.137	8.78	0.0778	0.0696	−10.52
	0.1218	1.3424	1.4769	10.02	0.0999	0.0904	−9.5
	0.2447	0.278	0.2942	5.83	0.0207	0.018	−12.95
	0.6617	0.1112	0.1138	2.34	0.0083	0.007	−15.82
	0.7789	0.0903	0.0981	8.64	0.0067	0.006	−10.64
	0.9641	0.0846	0.0916	8.27	0.0063	0.0056	−10.94
	1.1732	0.067	0.0759	13.28	0.005	0.0046	−6.82
	1.3325	0.0587	0.0664	13.12	0.0044	0.0041	−6.95
	1.4080	0.0521	0.0571	9.6	0.0039	0.0035	−9.85
	The Density of Micro-Pb and Nano-Pb were (0.0744 and 0.0612 g·cm^−3^)						

**Table 6 polymers-15-04416-t006:** The half value layer and mean free path of the pure, 10% and 50% lead/polymer composites as a function of photon energy.

Energy, MeV	HVL of Pure Lead, cm	HVL, cm					MFP, cm					MFP of Pure Lead, cm
		Pure	10% Micro	10% Nano	50% Micro	50% Nano	Pure	10% Micro	10% Nano	50% Micro	50% Nano	
0.0595	0.013	98.458	36.462	34.744	4.946	5.569	142.045	52.604	50.125	7.136	8.035	0.019
0.0810	0.030	109.329	54.665	56.722	8.910	10.929	157.729	78.864	81.833	12.855	15.768	0.043
0.1218	0.019	129.803	48.882	45.783	6.938	7.674	187.266	70.522	66.050	10.009	11.072	0.027
0.2447	0.105	157.892	137.803	134.071	33.502	38.530	227.790	198.807	193.424	48.333	55.586	0.152
0.3443	0.230	170.306	177.275	183.372	57.666	70.442	245.700	255.754	264.550	83.195	101.626	0.332
0.6617	0.591	214.597	179.572	180.978	83.713	99.590	309.598	259.067	261.097	120.773	143.678	0.853
0.7789	0.707	233.383	210.683	199.754	103.147	115.525	336.700	303.951	288.184	148.810	166.667	1.020
0.9641	0.870	270.761	213.934	203.867	110.023	123.776	390.625	308.642	294.118	158.730	178.571	1.255
1.1731	1.026	312.228	262.556	239.016	138.907	149.385	450.450	378.788	344.828	200.401	215.517	1.480
1.3330	1.123	325.421	296.217	269.707	158.615	170.726	469.484	427.350	389.105	228.833	246.305	1.620
1.4080	1.161	334.854	331.649	312.228	178.646	198.610	483.092	478.469	450.450	257.732	286.533	1.675

## Data Availability

All data generated or analyzed during this study are included in this published article.

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
