# Peer review of "Developing of Lead/Polyurethane Micro/Nano Composite for Nuclear Shielding Novel Supplies: γ-Spectroscopy and FLUKA Simulation Techniques"

_polymers, 2023, doi:10.3390/polym15224416_

Round 1

Reviewer 1 Report

Comments and Suggestions for Authors

Manuscript Title: Developing of Lead/Polyurethane Micro/Nano Composite for  Nuclear Shielding Novel Supplies: γ- Spectroscopy and FLUKA  Simulation Techniques.

Dear Editor of Polymers,

I would like to thank you for choosing me as a reviewer in your journal.

The manuscript “Developing of Lead/Polyurethane Micro/Nano Composite for  Nuclear Shielding Novel Supplies: γ- Spectroscopy and FLUKA Simulation Techniques” by El-Khatib A. et al. reports an original huge work about an the shielding abilities of polyurethane micro/nano composite, focused on he experimental efforts intended to compare the effectiveness of nano and micro particle size of Pb on attenuating polyurethane. The manuscript is fully suited to publication in Polymers Journal due to the originality and novelty of the topic covered. The manuscript is articuled in:

(i)              a first part on the detailed introduction to the aim of the work;

(ii)             a part of the methodologies adopted to fulfill the goal of the work by using the experimental and theoretical tasks “Developing of Lead/Polyurethane Micro/Nano Composite for  Nuclear Shielding Novel Supplies: γ- Spectroscopy and FLUKA  Simulation Techniques”;

(iii)          then, the results and discussion part where the data are analysed by a strong statistics. The abstract and conclusion sections fully hit the target and the results obtained in the work, and reflects the overall idea of the manuscript.

The paper is very well conceived, well organized and well written in adequate English language. The title is perfectly appropriate and of strong impact. The main goal of this work is clear and the activities done to fulfill the goal are very well described. The analysis on the data is clear and complete, using appropriate statistical methodology. The main output is clearly described and presented in a good form. The figures and captions are very clear and understandable.

The manuscript is fully suitable for the Journal, and perfectly falls into the aims of polymers. The topic is of interest to readers of the Journal. In conclusion, I recommend the manuscript to be accepted for publication after minor revision:

1-     Mnay previous studies are talked about the radiation shielding using polymers materials, so the authors can add some of these studies in the introduction section.

2-     In Materials and methods section: This section is very long. It’s better to describe this section indetail like a supplementary material.

3-     In Fluka section: I guess that the deviation between fluka and experimental are correct, please review it carfully. If possible send the justification between experimental and fluka results.

4-     In section “3.11. Neutron Macroscopic and Effective Removal Cross-Section Calculations “, it is not crrect to analyse theoretical. It is also neeed to justify the results experimentally. If you can not to justify it I advise the authors to remove this section from the manuscript.  

Comments on the Quality of English Language

good

Author Response

1-Many previous studies are talked about the radiation shielding using polymers materials, so the authors can add some of these studies in the introduction section.

References are added.

2- In Materials and methods section: This section is very long. It’s better to describe this section in detail like a supplementary material.

It has been corrected.

3-     In Fluka section: I guess that the deviation between fluka and experimental are correct, please review it carfully. If possible send the justification between experimental and fluka results.

      The slight nonconformity between the theoretical and experimental results has been the focus and interest of many researchers [34]. It can be due to the effect of collimator, size and hole, and the absorber thickness. For gamma rays, the accurate measurements of MAC (cm2 g-1) ideally need perfect narrow beam irradiation geometry. However, usually, the practical geometries used for the experimental investigations deviate from perfect-narrowness thereby the multiple scattered photons cause systematic differences in the measured values of MAC. Good agreement of theoretical and measured values of MAC was observed for all absorbers with thickness less than or equal to 0.5-1 mean free paths, thus considered it as optimum thickness for low-Z materials in the selected energy range. Some times it is very difficult to prepare samples with these specifications in order to satisfy these measurements. This leads us to use the available samples in sizes and volumes that we can suitable manufacture.

On the other hand, FLUKA uses different libraries BROND-3.1, ENDF-VIII0, JEFF-3.3, and JENDEL- 4.0. The photon interaction libraries contains records to define the interaction of photons with the elements Z = 1 - 100 over the energy range up to 100 MeV. These libraries has been designed to meet the  needs of users to model the interaction and transport of primary photons. It should be mentioned that using the default JENDEL library through the simulation process, may be one of the reasons for such deviations specially in the low-Z mode.

[49] Ref. Kulwinder Singh Mann, Manmohan Singh Heer, Asha Rani, Effect of low-Z absorber's thickness on gamma-ray shielding parame`ters, Nucl. Instrum. Methods Phys. Res. Sect. A 797 (2015) 19–28.

4-     In section “3.11. Neutron Macroscopic and Effective Removal Cross-Section Calculations “, it is not crrect to analyse theoretical. It is also neeed to justify the results experimentally. If you can not to justify it I advise the authors to remove this section from the manuscript. 

This part has been deleted as advised

Reviewer 2 Report

Comments and Suggestions for Authors

This work presents the investigation of the effect of adding Pb nanoparticles in polyurethane foams to improve thermo-physical and mechanical properties. The shielding efficiency of these foam samples has been studied by XCOM calculations and by using the FLUKA Monte Carlo code.

Recently, there have been many publications about materials of this type, with different compound contents. In this manuscript several testing methods were applied, however, it is not explained for what purpose. The authors used FLUKA simulations with a low number of histories, but photon results could be obtained with XCOM calculations alone.

Comments:

1. Can you explain the purpose of all the different testing methods?

 2. It is essential that the manuscript is well-structured and well-organized. Any spelling, grammar, or punctuation errors should be corrected. For example, the "3.8. Fluka Simulations" section should be placed in the "Methods" section. Section "2.4. Experimental" is empty. 

3. It is recommended to remove Table 3 as it is not necessary.

4. In Fig. 14 It is recommended to remove the MFP graph since it does not make any contribution.

5. Fig.4b is not clear.

6. It seems that the reference numbers have been misplaced or altered. Please correct it.

7.  Reference 14: please organize.

8. There is a mismatch in reference 18.

Comments on the Quality of English Language

Extensive editing of the English language is required.

Author Response

  1. Can you explain the purpose of all the different testing methods?

  The different characterization techniques describe the state of the dispersion of fillers in foam. The effects of these additions in the foam were evaluated, Fourier transform infrared spectroscopy (FTIR), scanning electron microscopy (SEM), transmission electron microscopy (TEM), and X-ray diffraction (XRD) have all been used to analyses the morphology and dispersion of lead in polyurethane. The findings demonstrate that lead is uniformly distributed throughout the polyurethane matrix. The compression test demonstrates that the inclusion of lead weakens the compression strength for the nanocomposites in comparison to that of pure polyurethane. The TGA study shows that the enhanced thermal stability is a result of the inclusion of fillers, especially nanofillers. The shielding efficiency has been studied, MAC, LAC, HVL,  MFP and Zeff  were determined either experimentally or by Monte Carlo calculations. The nuclear radiation shielding properties were simulated by the FLUKA code for the photon energy range of 0.0001–100 MeV.

  1. It is essential that the manuscript is well-structured and well-organized. Any spelling, grammar, or punctuation errors should be corrected. For example, the "3.8. Fluka Simulations" section should be placed in the "Methods" section. Section "2.4. Experimental" is empty. 

    This 3.8 part has been moved to Method section 2.2.10 as advised
  2. It is recommended to remove Table 3 as it is not necessary.

This part has been deleted as advised

  1. In Fig. 14 It is recommended to remove the MFP graph since it does not make any contribution.

This part has been removed as advised

  1. Fig.4b is not clear.

Clearification for Fig 4b has been done as advised

  1. It seems that the reference numbers have been misplaced or altered. Please correct it.

Reference corrected

  1. Reference 14: please organize.

Reference 14 corrected 

  1. There is a mismatch in reference 18.

It is corrected

Round 2

Reviewer 2 Report

Comments and Suggestions for Authors

As per the request, the authors have made the necessary corrections to the manuscript.